# OpenReview forum: "AI Der Ring:The Forging of the Future A Wagnerian Gesamtkunstwerk Reimagined through Multi-agent System"
_ICLR.cc/2026/Conference — Submitted to ICLR 2026_

### Official Review · Reviewer_aZkM · 2025-10-24

**Soundness:** 1
**Presentation:** 2
**Contribution:** 2
**Rating:** 2
**Confidence:** 3

**Summary:**

This paper introduces the “AI Der Ring” framework, which applies multi-agent systems to opera creation, enabling AI agents to collaborate in composing music, designing stages, performing multiple roles, and interacting with live audiences.
Using the YuLan-WanXiang social simulation platform, the authors simulate AI-driven operatic productions to test narrative coherence, adaptability, and audience engagement. The paper argues that multi-agent AI can act as a co-creator in large-scale performative art.

**Strengths:**

The paper presents an interesting fusion of AI and the arts. This area is arguably quite understudied, and it is refreshing to see work in this direction. It presents an interesting idea of combining operatic art, multiagent systems, and Wagnerian aesthetics.

**Weaknesses:**

This paper has several weaknesses and oversights. Some more work on the points mentioned below could strengthen the paper.

The discussion on this particular multi-agent system only covers it conceptually. There isn’t any technical/architectural details in regard to the implementation of the multi-agent system. The methods seemingly rely on general claims. For example, “The agents communicate via a well-defined protocol that allows them to exchange ideas, provide feedback, and collectively make decisions about the narrative direction.” What protocol? And how do agents interact through it?

There also is no information regarding the evaluation framework. The scores listed in Table 2 are not defined and ungrounded. Were they human ratings or automated? Were the scores defined on a 1-5 scale? What was the sample size? What was the methodology behind the evaluation?

Furthermore, the use of the YuLan-WanXiang platform is only mentioned in name; the details of the evaluation through the platform are not described in-depth. As such, the claims and conclusions from the paper are completely unreproducible, which limits its potential impact on the community.

The formatting is also quite underdeveloped. The title of the paper seems to have been unchanged from the formatting document. There are also no citations throughout the text that refer to the references. Models like Magenta, Musenet, and Stable Diffusion are mentioned but not cited. Several references to previous work are not cited. For example, in Sec. 2.5, mentions of “Early experiments with multi-agent systems in theater and dance have demonstrated their potential [...]”.

**Questions:**

To address the points above, it’d be great to get some answers from the author(s) on the following questions:

How are the multi-agent systems technically implemented?

How is coordination among agents achieved in generating the narrative, music, and visual elements?

The paper references a “shared knowledge base.” What is its structure? How do agents interact with it?

What metrics or criteria were used to generate the quantitative evaluation scores in Tables 1 and 2? Who assigned these scores (human experts, participants, or the system itself)?

What were the computational resources and setup required to run the multi-agent simulation on the YuLan-WanXiang platform?

---

### Official Review · Reviewer_eKd3 · 2025-11-02

**Soundness:** 1
**Presentation:** 1
**Contribution:** 1
**Rating:** 0
**Confidence:** 4

**Summary:**

This paper presents a framework for reinterpreting Richard Wagner’s Der Ring des Nibelungen as a computational Gesamtkunstwerk through multi-agent systems. It argues that AI can act as both a tool and a collaborator in operatic creation. The proposed system includes Multi-agent collaborative narrative construction, Single-agent multi-role performance, Dynamic scene adaptation, Interactive audience engagement, and Integration of historical and cultural context. A simulation was reportedly implemented, testing narrative coherence, audience engagement, and technical efficiency. Tables rate these aspects highly and the conclusion envisions AI as a generative co-creator for future Wagnerian reinterpretations.

**Strengths:**

The paper tries to merge art history, philosophy, and multi-agent AI in a creative way, and raises valid questions about authorship, human–AI co-creation, and ethics in artistic production, which seems to be an interesting topic.

**Weaknesses:**

The mechanical and LLM-like writing style makes me believe that an extensive amount of body is generated by LLM. Some suspicious evidence:

1. Repetitive phrasing (e.g., “multi-agent system” appears over 100 times, often with identical syntax errors like missing plural s).
2. Overuse of transitional clichés like "not merely…but,” “represents a pivotal evolution,” “comprehensive framework”) typical of generative models.
3. Sentences have balanced clauses, excessive synonymic variation...
4. No citations from musicology, performance studies, or primary Wagner sources, and only generic AI papers and arXiv preprints, which is inconsistent with a genuine interdisciplinary project.

In fact, I use open tools to test the likelihood that this paper is generated by LLM and the result is over 90%. Due to this concern, I did not spent additional time verifying the intellectual merit of this paper and will report an ethical concern.

**Questions:**

No question for this paper.

**Details Of Ethics Concerns:**

I believe this paper is written by LLM.

---

### Official Review · Reviewer_PvoU · 2025-11-03

**Soundness:** 1
**Presentation:** 1
**Contribution:** 1
**Rating:** 0
**Confidence:** 5

**Summary:**

This paper presents the "AI Der Ring" framework, a concept for reimagining Richard Wagner's opera cycle "Der Ring des Nibelungen" as a computational "total work of art" using artificial intelligence. This framework is built on four key dimensions: multi-agent collaborative narrative construction, single-agent multi-role performance, dynamic scene adaptation, and interactive audience engagement. The system was experimentally validated using the YuLan-WanXiang simulation platform, which confirmed its stability, narrative coherence, and responsiveness. Evaluations demonstrated strong performance in areas like narrative collaboration and role-playing adaptability, achieving high scores for audience engagement, although limitations related to computational load and multimodal synchronization were identified. The researchers plan to use this foundation to generate four new multi-agent operas reinterpreting the entire Ring cycle, aiming to bridge historical art with AI-driven creativity and explore the resulting ethical questions of authorship.

**Strengths:**

Exploring the integration of AI with artistic creation is a relatively novel question, especially the art form of opera.

**Weaknesses:**

The most severe problem in this article is, it appears to be unfinished; in particular, a compilation bug in the title has left the original template’s placeholder in place.

The writing in this article is extremely poor, with numerous typos, compilation bugs, and telltale signs of AI-generated content.

The experiments in this paper do not answer the research questions raised in the introduction, what challenges arise when applying AI to opera composition, how are they addressed, and how well are they addressed, and there is a severe lack of evaluation by human experts.

The article overlooks the most important component of opera creation: the music. It also doesn't explain why only Der Ring des Nibelungen was chosen as the object of study.

The amount of information presented in this article is very limited, with a serious lack of reproducible evidence and insufficient examples and supplementary materials to demonstrate the results of the opera created.

**Questions:**

Der Ring des Nibelungen should have been composed between 1848 and 1874, not 1843 to 1874 as stated in the article?

---

### Official Review · Reviewer_h9kB · 2025-11-03

**Soundness:** 1
**Presentation:** 3
**Contribution:** 1
**Rating:** 0
**Confidence:** 5

**Summary:**

The manuscript has been desk rejected due to a violation of the double-blind review policy. Specifically, the paper includes explicit institutional information: "Gaoling School of Artificial Intelligence at Renmin University of China." This detail compromises the anonymity required for a fair and unbiased review process.

**Strengths:**

See in Summary

**Weaknesses:**

See in Summary

**Questions:**

See in Summary

---

### Official Review · Reviewer_ZEVw · 2025-11-05

**Soundness:** 1
**Presentation:** 2
**Contribution:** 2
**Rating:** 2
**Confidence:** 4

**Summary:**

An attempt at creating an AI-driven re-interpretation, or re-enactment, of Wagner's Nibelungen opera. The various roles are acted by AI agents, there are are also other agents taking the role of AI directors, and there is an audience engagement feature.

**Strengths:**

This was a fascinating and unexpected paper. Much more interesting to read than most of the soulless write-ups of minor algorithmic variations you typically see in the review pile for a conference such as this. I would like to see, or experience rather, this gesamtkunstwerk. And I want to know how it actually works in detail. I wanted to like the paper.

**Weaknesses:**

This is a technical conference and we expect to see papers that describe their technical contributions in detail and describe them rigorously. Alas, this paper is alarmingly scarce on such details. There is simply no insight into how these agents actually work. You never open the lid and tell us what kind of model is used, what input it takes, how often it is called, what the output looks like, and so on. The system diagram you present is so vague as to not really tell us anything at all. How many agents are there, which exact roles do they play, how do they communicate? The claims about how well the system works are also unsubstantiated; you can't just present a number ("4.6"), you need to present a whole evaluation methodology.

I realize that the whole system as described would make for a very long writeup where you to write it up at the required level of detail. So, for submitting to a technical conference, you would need to focus on some particular subsystem or aspect of the whole system.

You would also need some kind of hypothesis or research question. Reading the paper, I kept wondering why. Why did you this, not that, etc.

Finally, there is an almost complete lack of citations, except for apparent self-citations.

Sorry for sounding so negative. The work you guys did seems really cool, it's just not written up in the way papers are supposed to be written at technical conferences.

**Questions:**

I really want to know how the agents actually work, but I don't think that's something you will have space and time to answer within the limits of a rebuttal process. I think you need to write a new paper if you want to publish this work at a technical conference.

---

### Meta-Review · Area_Chair_GctW · 2026-01-06

**Summary:**

All rejections and no rebuttal.

**Reviewer Concerns:**

Limited Novely.

**Reviewer Scores:**

NA

---

### Decision · Program_Chairs · 2026-01-26

Reject